# Assessment of the Factors Influencing the Performance of the Adoption of Green Logistics in Urban Tourism in Thailand's Eastern Economic Corridor

**Sanhakot Vithayaporn** [1,*] , **Vilas Nitivattananon** [1] , **Nophea Sasaki** [1] and **Djoen San Santoso** [2]

1 Department of Development and Sustainability, Asian Institute of Technology, Pathum Thani 12120, Thailand
2 Department of Civil and Infrastructure Engineering, Asian Institute of Technology, Pathum Thani 12120, Thailand
* Correspondence: sanhakot@hotmail.com

**Abstract:** Tourism plays a crucial role in promoting economic growth, but it can also contribute significantly to environmental degradation, particularly in urban areas where there is a high concentration of local residents and visitors. Tourism is crucial for economic development but can also harm the environment, particularly in urban areas where both locals and tourists are concentrated. Adopting green logistics is important for promoting sustainable urban tourism while minimizing environmental impact. However, little research has been conducted on this topic in Thailand. This study aimed to identify the factors that influence the performance of green logistics in urban tourism activities in Thailand's Eastern Economic Corridor (EEC). Semi-structured interviews were conducted with 25 leading logistics enterprises and five major factors were identified: The implementation of a green transportation system, the level of the environmental management system, the enhancement of reverse logistics, the level of government governance, and the perceived usefulness of green logistics for logistics enterprises. The research found that both the government and enterprises play a key role in initiating green logistics, and this action is the mechanism behind the identified factors. The study's holistic perspective on the contributions of green logistics to urban tourism has academic implications and can inform decisions on enhancing and improving green logistics performance for sustainable regional development. The study concludes with implications and recommendations for future research.

**Keywords:** green logistics performance; green logistics enterprise; sustainable development; urban tourism

## 1. Introduction

The global tourism industry has been significantly impacted by the outbreak of COVID-19, resulting in worldwide lockdowns and a near collapse of the industry. However, as the outbreak has gradually been contained, the tourism industry is starting to recover, especially in Thailand, where the government has implemented relaxed policies to open the country to tourists. The United Nations has predicted that by 2030, 60 percent of the world's population will live in urban areas, up from 54 percent in 2015 (UNWTO 2019). This, coupled with the increasing concentration of greenhouse gases in the atmosphere and rapidly rising urban populations, has led to a shift in the tourism development paradigm towards new practices, particularly in urban tourism development. Surprisingly, between 2004 and 2017, the urban tourism sector had the most rapid growth in the world, making it a key factor in supporting economic development (Nilsson 2020). Therefore, an effective logistics operations system is needed to facilitate the flow of tourists to cities, from city to city, and beyond, particularly in urban areas.

Postma and Schmuecker (2017) stated that tourism, as an essential industry, is also a major contributor to environmental damage and $CO_2$ emissions, particularly in cities with

urban tourism. Likewise, the vigorous development of logistics plays a significant role in driving economic growth (Yang et al. 2019; Rashidi and Cullinane 2019) as evidenced by the sharp increase in logistics enterprises in various fields including tourism over the past several years. Nevertheless, the range of negative impacts on the environment caused by logistics development has become increasingly prominent in the forms of pollution, noise, waste, and major emissions of GHG produced by fuel consumption (Wang et al. 2018). These impacts have been accelerated by the increasing number of tourist arrivals in the main cities across Thailand. Worse yet, as urban growth is very dynamic, the sharp increase in urban tourists can lead to the worsening of the already deteriorated environmental conditions in urban areas. Therefore, green logistics have gained attention in the field of urban tourism as a tool to reduce such impacts while ensuring that urban tourism development can still be pursued (Dabphet et al. 2012). One previous study suggests that green logistics development is a solution to control environmental impacts and mitigate climate change (Khan et al. 2018). The development of green logistics is also aligned with the UN's Sustainable Development Goals (SDGs), which aim to increase awareness of the organic combination of social well-being, economic growth, and environmental resource protection. Based on the current environmental pollution status of tourism and logistics, green logistics activities need to be aligned with sustainable development because it results in improved development, particularly within the regional context.

Accordingly, logistics enterprises involved in urban tourism activities need to focus on greener goals for their business through the adoption of green logistics in the Eastern Economic Corridor (ECC) of Thailand. Previous research has shown that managing the flow of people through integrated green logistics principles plays a vital role in urban tourism. Green logistics is a crucial factor in the development of sustainable tourism as it enables tourists to access their destinations while minimizing negative environmental impacts. According to Muangpan and Suthiwartnarueput (2019) and McKinnon (2021), logistics within the tourism industry includes transportation and various components such as carriers, accommodations, restaurants, tourist attractions, car rental services, and overall ambience, which encompasses the appearance and punctuality of the service provided. To ensure a seamless experience, it is crucial to have collaboration and coordination among the different activities and areas involved. Therefore, it is essential to enhance the performance of green logistics towards sustainable tourism development, especially in urban tourism, where the majority of tourism activities take place. The key principles of green logistics performance include green design, green transportation, environmental management systems, and reverse logistics, as identified by (Dabphet et al. 2012).

While ecotourism and nature-based tourism are often the focus of sustainable tourism development, it is important to recognize that the majority of tourism takes place in urban areas where effective green logistics are essential. This study seeks to address the gap in the literature by examining strategies for implementing green logistics in the Eastern Economic Corridor (EEC) region of Thailand. The EEC is a flagship project of the Thai government and a major tourist destination, with 28.89 million international arrivals in 2019. However, with a population of approximately 29 million, the region is heavily congested. The development plan for the EEC emphasizes the importance of effective green logistics performance as a key component of sustainable tourism growth.

Effective green logistics performance is the ultimate goal of the EEC development plan for this regional gateway of tourism growth. Logistics enterprises have played a significant role in adopting green logistics for urban tourism. However, there is still a gap, as many logistics enterprises are not focused on green logistics. To address this gap, this study aims to assess the factors influencing the performance of the adoption of green logistics in urban tourism in the EEC area of Thailand, which could form a basis for better-informed decision-making toward sustainable development.

## 2. Literature Review

The literature suggests that green logistics performance encompasses several aspects, which can be categorized into several research directions. These include green logistics in urban tourism, environmental conservation and emissions reduction through green logistics, and the establishment and evaluation of the current performance of green logistics in logistics companies operating in urban tourism, all contributing to the achievement of sustainable development.

### 2.1. Green Logistics from an Urban Tourism Perspective

The interlink between green logistics and urban tourism is clear and can be seen in the green transport-related activities and the entire infrastructure that plays a role in the accessibility of tourists to a certain destination, which has a direct impact on the efficiency of sustainable tourism, as well as fuel-efficient vehicles, biofuels, electric vehicles (EVs), bicycles, and tricycles for delivery (Rashidi and Cullinane 2019). Other concerns are the consumption and production of green logistics in urban tourism areas, including accommodation, restaurants, and sightseeing in the entire package provided by the enterprises to tourists. Logistics has been defined with various meanings depending on the context and perspective. (Bracken 2022) defined logistics as a science dealing with the integrated management of all the materials and the corresponding information flow from suppliers through the transformation of input materials to the end consumer. However, logistics can also be defined based on diverse industries. As mentioned in the introduction, logistics in the tourism industry involves all of the kinds of activities that tourists consume during their trips. (Grah et al. 2020) define green logistics as all attempts to minimize the ecological impact of logistics activities with all kinds of logistic systems. Perkumienė et al. (2020) stated that the term "green logistics" is often used interchangeably with "reverse logistics," but in contrast to reverse logistics, green logistics "summarizes the logistics activities that are primarily motivated by the environmental considerations." Thus, Dente and Tavasszy (2018) have defined it as "the integrated management of all the activities necessary to mobilize products through the supply chain considering environmental and sustainable issues." Therefore, green logistics from a tourism perspective involves government agencies, the private sector, and enterprises concerned with the development of the tourism industry, and green logistics initiatives are necessary for tourism areas, particularly urban tourism where tourist arrivals are the most prominent.

Urban tourism accounted for the place where tourists spent a night for almost two-thirds of the total number of nights spent per trip (Ibnou-Laaroussi et al. 2020). Tourism statistics show increasing urban tourism and higher occupancy rates in urban areas compared to rural areas and peri-urban tourism. Urban tourism is different from other forms of tourism in that tourists normally travel to places with a high population density, and then the time spent at that place is generally shorter than is normally spent on leisure vacations. The United Nations World Tourism Organization (UNWTO) defined urban tourism as "a tourist activity in an urban area, while city/urban destinations provide a wide range and variety of attractions in cultural, architectural, technological, social and natural experiences and products for leisure and business." Urban tourism planning and management require the equality of the logistics system to develop at the same time with the conservation of the environment, culture, and society, as well as transportation within the urban areas, to support sustainable development (Han 2021). Green logistics performance is the solution to the problems of urban tourism development. Stakeholders such as enterprises involved in urban tourism play a crucial role in improving the performance of green logistics.

### 2.2. Environmental Conservation and Emissions Reduction by Applying Green Logistics to Urban Tourism

The conservation of the environment and the reduction of emissions in logistics systems have been shown to significantly slow down climate change and global warming caused by the use of fossil fuels in logistics-related transportation and consumption-

production activities (Larina et al. 2021). However, the development of urban tourism heavily relies on logistics systems to serve the flow of tourists during their stay, as mentioned above. Therefore, policies and measures for green logistics performance should be implemented to achieve sustainable development (Anser et al. 2020). Effective measurement of green logistics performance should be applied to protect the environment, reduce emissions, and improve the vitality of urban tourism traffic flow and tourist consumption during their stay. In the context of climate change mitigation, reducing $CO_2$ emissions associated with the logistics process in urban tourism is a major challenge for governments. Achieving emission reduction goals is influenced by two aspects. First, from the perspective of transportation mode, the reduction effect of emissions is greater for railways and waterways than for highways and aviation. Second, government regulations play a role. The government implements environmental policies and interacts with logistics enterprises to encourage energy conservation and emissions reduction behaviors. Therefore, carbon-pricing regulations have been implemented for tourism enterprises (Jantakat et al. 2021). Consumers in the tourism industry perspective are tourists who consume all kinds of logistics activities; hence, protecting the environment and emissions reduction should be considered for enterprises in the tourism industry. Thus, this study specifically focuses on urban tourism in the EEC, Thailand, by collaborating to launch campaigns of green logistics performance in the destinations provided to the tourists. The logistics enterprises involved in tourism destinations should initiate the promotion of environmental awareness through their various campaigns for green initiatives, for example, hotel enterprises, restaurant enterprises, and transportation enterprises, and the adoption of green practices in the hospitality sector, which will enhance their green logistics performance at the destination. Therefore, the green logistics concept has been promoted from an urban tourism perspective to reduce carbon emissions that result from tourists and tourism enterprises to achieve sustainable development goals.

*2.3. Green Logistics in Urban Tourism Contributes to Sustainable Development*

The contribution of the logistics sector considerably impacts the growth of the country's economy, which involves several sectors, such as transportation, consumption, and production in the general industries related to the logistics sector. However, logistics from the tourism perspective concerns transportation, accommodation, restaurants, attractions at tourist destinations, etc. The UNWTO defines sustainable tourism as "tourism that yields current and future economic, environmental, and social impacts on the needs of the industry, visitors, host communities, and the environment." Thereby, the "green" or "sustainable" concept advocates for minimal detrimental impacts on the local environment and the promotion of cultural benefits in the region for its local inhabitants in the social pillar of the SDGs. Since the declaration of the United Nations in 2017 of the International Year of Sustainable Tourism, this sector has expanded. Moreover, the global concept of the Bio-Circular-Green (BCG) Economy model leads to an advanced national economy, a clean environment, and social well-being as a tool to achieve the SDGs.

The concept of green logistics emerged as a vital and integral part of the firms' efforts to act in environmentally responsible ways and contribute to the economy and society. Several researchers have explored the association between green logistics and sustainable development. Likewise, McKinnon (2021) stated that the logistics performance index (LPI) is one of the indicators to assess the performance of logistics. According to the logistics performance index (LPI) certified by the world bank in 2018, Thailand is ranked no.34 out of 167, with the highest performance of 80.20 percent reached (Li et al. 2021). This number indicates the efficient operation of logistics, which means a higher efficiency is a higher impact on the environment. Hence, this reinforces that logistics performance in Thailand is necessary for the transformation into green logistics. In the context of the growing concerns about environmental issues caused by the logistics sector, reporting the results of sustainability initiatives has become an important mechanism to appease stakeholders' concerns. The Global Reporting Initiative (GRI) was proposed as a framework to measure

whether the logistics sector is environmentally friendly. The purpose of the GRI framework is to provide detailed guidance on the disclosure of specific indicators of corporate social responsibility (CSR) performance (Yang et al. 2022). Given the increase in global awareness of the consumption and production of commodities across countries, environmental issues have become major concerns for logistics enterprises (Chhabra et al. 2022). Consequently, green logistics has emerged as a concept that includes a set of green activities minimizing the total environmental impact of logistics enterprises (Magazzino et al. 2021) and providing environmental protection and sustainability. The negative impacts of the components of tourism logistics such as transportation, hotels, urban development, and restaurants on the environment should also be mentioned. Moreover, urban tourism should be sustainably developed to fulfil tourists' right to a clean environment. Sustainable urban tourism development is not an isolated or special form of tourism; rather, it should be contributed to by the use of green logistics performance.

### 2.4. Green Logistics Performance

Green logistics performance refers to all the activities involved in distributing finished products to end consumers, ensuring they arrive at the right place and time. In the context of tourism logistics, the focus is primarily on transportation modes that facilitate the movement of tourists, such as airplanes, trains, buses, cars, and motorcycles, as well as related services such as hotels, restaurants, attractions, and shops, which are major contributors to greenhouse gas (GHG) emissions and air pollution. To illustrate, optimizing vehicle routing and scheduling to transport tourists, or increasing vehicle utilization, can significantly reduce the environmental impact of logistics enterprises. This approach achieves resource conservation (i.e., energy and fuel) and lowers carbon emissions.

Green logistics performance encompasses all of the activities that distribute completed products to end consumers at the proper time and place. Logistics in tourism is primarily concerned with transportation to facilitate the flow of tourists, such as airlines, trains, buses, cars, and motorcycles, followed by hotels, restaurants, attractions, and shops, which largely contribute to GHG emissions and air pollution. For instance, the optimization of vehicle routing and scheduling to move tourists or an increase in vehicle utilization reduces the environmental impact of logistics enterprises as a result of reducing their consumption of resources (i.e., energy and fuel) and their carbon emissions (Crellin et al. 2022).

In past research, scholars (Gazzola et al. 2019; Ministry of Energy 2022; Mironescu et al. 2021) studied green logistics performance in various industries using the Logistic Performance Index (LPI) published by the World Bank as a guideline for green logistics performance measurement. This study adopts the indicators of LPI to assess the logistics of tourism in terms of green logistic performance; hence, the assessment of logistics enterprises from the urban tourism perspective was constructed. Secondary data were obtained by LPI, the national logistics association, and reports from logistics enterprises in the urban tourism industry, whereas the primary data to assess green logistic performance from an urban tourism perspective was obtained from the semi-structured interviews with logistics enterprises (Wang et al. 2018).

## 3. Methodology

Based on the literature, the qualitative method of semi-structured interviews was conducted face-to-face, via Zoom conferences, and on the telephone, with logistics enterprise considered an appropriate approach to reach the assessment factor of green logistics performance in urban tourism contributing to the sustainable development of the logistics enterprises in the Eastern Economic Corridor of Thailand. Thus, the data were recorded by memo and voice recordings to transcribe them for data analysis.

### 3.1. Research Design and Overall Methodology

The problem statement and rationale of the study are as follows: Urban tourism areas are threatened by tourists consuming logistics activities during their stay, which impacts

the environment and increases carbon emissions. Moreover, there is an incapacity of local governments to manage green logistics. The involvement synergy concept is considered an important mechanism to address the threats to urban tourism and beneficial for multiple stakeholders of logistics enterprises in contributing to sustainable development. The conceptual framework of this study is shown and explained in Figure 1.

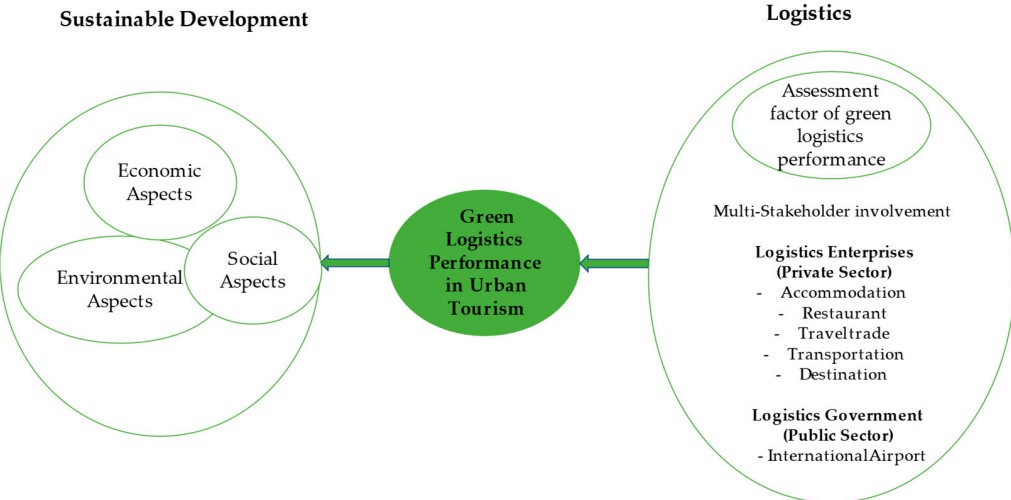

**Figure 1.** Conceptual framework of assessment of the factors influencing performance of the adoption of green logistics in urban tourism. Source: Adopted from Jotaworn and Nitivattananon (2023).

Based on the conceptual framework, the qualitative method was used to explain the assessment factors of green logistics performance in urban tourism that contribute to sustainable development. What is the derived benefit from synergy opportunities of green logistics performance? Hence, semi-structured interviews were used to obtain a narrative from the airport authority as the public sector and relevant stakeholders from the private sector (accommodation, travel trade, restaurants, destinations, and transportation). The methodology included four steps:

Step 1: Qualitatively exploring the review of the selected secondary data through related literature and organizational reports such as the EEC office to find the potential and challenge factors. This step also included a study area, population, and sampling size determination.

Step 2: Assessing the relevant stakeholders qualitatively through semi-structured interviews based on the interview questions to assess the factors that influence the green logistics performance of the logistics enterprises in urban tourism that contribute to sustainable development.

Step 3: Analyzing the synergistic opportunities, after which content analysis of a qualitative structure (coding, axial, and selective methods) was used to synthesize the result.

Step 4: Consolidating the findings from those results to identify factors based on green logistics performance.

*3.2. Study Area*

The study area covers Thailand's Eastern Economic Corridor (EEC) region (Table 1 and Figure 2). The EEC is a diverse area that offers tourists and visitors a wide range of attractions. The region consists of three provinces (Chachoengsao, Chonburi, and Rayong), and the region contributes 20 percent to the country's GDP and is a popular tourist destination with several famous attractions in the urban area. Each province has its characteristics in terms of urban tourism perspectives and involvement in logistics enterprises. Green logistics performance in the study area involves transportation, an international airport, accommodation, restaurants, and tourist attraction destinations. An

example of a green strategy in this study area is the EEC office's green policy, which includes all sectors of the economy, including tourism. Another representative from U-Ta-Pao International Airport has established a green space concept, ecological restoration, community environment, and a water retention policy utilizing a green approach to start green logistics.

**Table 1.** Study area profile.

| Provinces | Area (sq. km) | Population (People) | Density (per sq. km) | Tourist Arrivals | $CO_2$ Emissions Rate/Tons |
|---|---|---|---|---|---|
| Chachoengsao | 5351 | 785,973 | 146.88 | 460,429 | 1698 |
| Chonburi | 4363 | 1,567,000 | 359.15 | 908,954 | 3354 |
| Rayong | 3552 | 908,778 | 255.84 | 467,997 | 1762 |

Source: Eastern Economic Corridor Office of Thailand (EEC) (2022); Ministry of Energy (2022).

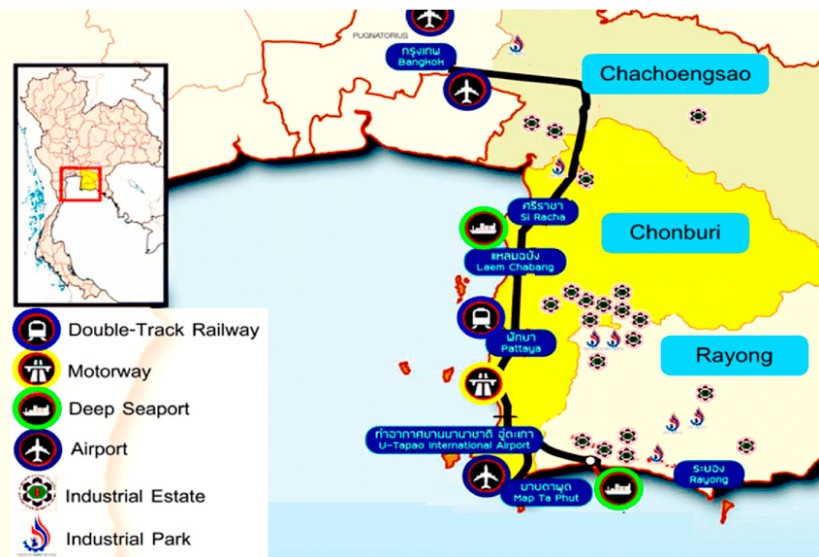

**Figure 2.** Map of the study area. Source: Eastern Economic Corridor Office of Thailand (EEC) (2022).

*3.3. Data Collection*

In the context of empirical studies on the assessment factors influencing green logistics performance, particularly from an urban tourism perspective, this study utilizes secondary data, the logistics performance index, logistics reports from the Eastern Economic Corridor Office, and the literature. Primary data collection employed a semi-structured questionnaire to conduct semi-structured interviews with participants to obtain first-hand data and explore the assessment factors of green logistics performance.

To define the sampling size in qualitative research, Creswell and Creswell (2017) suggested that the number of interviews should be between 20 to 30 for reliable results. Given the actual circumstances, 25 respondents in the green logistics enterprises in the urban tourism industry in the Eastern Economic Corridor region of Thailand were proposed as the unit of analysis for this study. Because the assessment of green logistics performance is influenced by the government and enterprises, respondents from the relevant government departments and logistics enterprises, which comprise both department heads and enterprise owners of both genders, were selected to obtain more comprehensive information (Table 2).

**Table 2.** Information of key informants (logistics enterprises in urban tourism as a unit of analysis).

| Province | Government/Logistics Enterprise | Position | Gender | No. |
|---|---|---|---|---|
| Chachoengsao | Hotel | Owner | Male | 2 |
| Chonburi | Private transport | Owner | Male | 3 |
| | Hotel | HR Director | Female | 2 |
| | Restaurant | Owner | Male/Female | 4 |
| | Travel agency | Owner | Male | 2 |
| Rayong | International Airport | Deputy Director | Male | 2 |
| | Government bureau | Director | Male | 1 |
| | Private transport | Owner | Male | 2 |
| | Hotel | GM | Male | 1 |
| | Restaurant | Owner | Male/Female | 4 |
| | Destination | Manager | Male | 2 |
| | Total | | | 25 |

For the secondary data of this study, the logistics performance index (LPI) indicators were adopted from recent related studies, whereas individual semi-structured interviews were conducted for the primary data collection via face-to-face Zoom meetings or video calls, depending on the situation of each interviewee, without interface with other respondents so that each person could speak freely and reveal more information. The semi-structured interviews focused on the following questions to achieve the research objectives of this study: 1. Do you know about the current green logistics concept? 2. Do you think the current green logistics concept is well known in Thailand's EEC region or nationwide? 3. Have you initiated any green logistics policies/campaigns in your department/company for tourist arrivals in the EEC area? 4. What is the current performance of green logistics in the EEC area in the urban tourism sector? 5. Do logistics operations in urban tourism areas in the EEC impact the environment and produce carbon emissions? 6. Is the Bio-Circular-Green Economy model (BCG) an effective tool for green logistics in urban tourism? 7. What should the government do to improve green logistics performance in the EEC region from an urban tourism perspective? 8. How can green logistics contribute to sustainable urban tourism development? 9 What are the challenges and opportunities to improve green logistics performance in urban tourism in the EEC area? Data collection commenced on 1 May 2022 and was completed on 31 August 2022. The study followed the steps of the qualitative method.

*3.4. Data Analysis*

Content analysis, a qualitative method, is a broad technique that seeks to provide a systematic and objective means of making valid inferences from verbal, visual, or written data to describe and quantify specific phenomena (Creswell and Creswell 2017). Therefore, the content analysis method was used in this qualitative study to analyze the data. The study used key qualitative structural methods: (1) Open coding, (2) axial coding, and (3) selective coding.

- The first analysis method, open coding, is the decomposition of collected data into manageable and analyzable segments. Through continuous abstraction, comparison, and brainstorming, the concepts that can represent the essence of those segments are extracted, and further categories are formed. The categories were identified from both the secondary and primary data, multiple rounds of data analysis, and synthesis to achieve the research objective.
- The second method is based on open coding and is called axial coding, which was performed to re-analyze the content and extracted categories, as well as the internal relations among these categories, divided into main categories.

- In the third method, after the open and axial coding, selective encoding was performed to explore the internal connections between the various main categories and extract them into a storyline, to fulfil the research objective. The data, categories, and main categories were analyzed and compared again.
- Finally, a theoretical saturation test was conducted to ensure the validity of the data through the data triangulation method. Data triangulation was conducted with five selected interviewees from each area consisting of respondents from the international airport, a hotel, a restaurant, a destination, and a private transportation enterprise, for follow-up feedback at various times. Moreover, two academic experts were proposed to be the investigators for the triangulation saturation test.

## 4. Results

This section presents the findings of the study, which are aligned with the scope and process of the research methodology. This includes exploring secondary data, assessing the primary data through semi-structured interviews, analyzing the synergies, opportunities, and challenges through the primary data, and finally, discussing and concluding the data.

### 4.1. Exploring the Data Qualitatively

The secondary data for this study were obtained from various sources, including related literature and organizational reports, such as the EEC Office. The study focused on the Eastern Economic Corridor (EEC) region, which aims to become an environmentally friendly region for its residents and visitors. To achieve this, the EECO has initiated several measures to minimize the environmental impact and carbon emissions from various activities in the area, particularly in the green logistics system. The study identified several key factors that influence green logistics, including the adoption of green design concepts in logistics, environmental concerns caused by logistics activities in tourism areas, the use of transportation and other vehicles in logistics, the implementation of reverse logistics initiatives, and the use of the BCG economy model to transform logistics into green logistics for business sustainability. Table 3 provides an analysis and synthesis of these key factors.

**Table 3.** Exploring secondary data of green logistics.

| A Key Factor of Green Logistic Performance | Statements |
| --- | --- |
| Greener goals and green design | The concept of going green refers to fostering development to ensure that natural assets continue to provide the resources and environmental services for the well-being of humans (Testa et al. 2021). The greening policy, green concept, and going green campaigns aim to achieve all of the SDGs in many aspects. The term "green" was developed in the 1980s as a business concept to use environmental issues for competitive advantages and then evolved into the modern and fashionable "green paradigm." Green packaging (GP) has its place. This is an "environmentally friendly package, which is completely made from natural plants, that can be recycled or reused, is prone to degradation and promotes sustainable development, it is harmless to the environment as well as to human and livestock health." Green logistics is influenced by the green design concept (Zhang et al. 2020). |
| The environmental impact caused by logistic activity in urban tourism | Tourism, as one of the essential industries, is also one of the major causes of environmental impacts and $CO_2$ emissions, particularly in cities with urban tourism (Testa et al. 2021; Meiksin 2020). The EEC Office and the Ministry of Energy in Thailand have documented that the total rate of carbon emissions in these three provinces reached up to 6814 tons (Gillingham and Stock 2018; Khdair et al. 2021). |

**Table 3.** *Cont.*

| A Key Factor of Green Logistic Performance | Statements |
| --- | --- |
| Transportation and Electric Vehicles options | Eco-labels in the transportation sector may prove to be effective tools for moving the industry toward decreasing GHG emissions and the overall impact on natural resources. Bicycle and walkability tourism support the green transportation concept. Logistics and transport-related factors are interlinked, as transportation is an active part of logistics, and green logistics includes transportation planning (Larina et al. 2021; Hamurcu and Eren 2020). |
| The BCG Economy Model to support green logistics performance | Pourmehdi et al. (2022) stated that Europe and Thailand have adopted the Bio-Circular-Green (BCG) Economy Model to lead to a highly developed national economy. The BCG includes the production of renewable biological resources and the conversion of these resources and waste streams into value-added products, such as food, feed, bio-based products, and bioenergy |
| Reverse logistics | Reverse logistics (RL) interacts with recycling, reusing, and reducing activities. The use of energy-saving equipment results in minimal impacts on the environment. Environmentally friendly materials can be used in the construction of warehouses with the use of renewable energy sources. By effectively applying reverse logistics, green logistics practices minimize their eco-impacts (Pourmehdi et al. 2022) |
| Business sustainability | Business sustainability refers to achieving an organization's vision and mission. It can be described as the application of knowledge, skills, tools, and techniques to the organization's activities, products, and services (Gillingham and Stock 2018). |

### 4.2. Assessing through the Semi-Structured Interview

The second step involved assessing the research questions through semi-structured interviews. Primary data were derived from 15 interview sessions with 25 interviewees, each lasting between 45 min to an hour. Some respondents provided repetitive answers, which challenged them to answer the questions. After each interview, the contents were sorted, and the material was immediately written in a memo. A total of approximately 6500 words of interview records were obtained. For two-thirds of the interview records (15 interviews, approximately 4200 words), the data were randomly selected for coding to explore the factors that influence green logistics performance. Axial and selective coding methods were used in the second step of assessing the green logistics performance.

### 4.3. Analyzing the Data

The results were obtained from the first step of analyzing secondary and primary data, which followed a qualitative structure. In the second step, interviews were conducted with 25 multi-key informants, and the results were extracted into 13 categories, with original statements from the interviews being identified and predicting the factors that influence green logistics performance through open coding analysis. The categories and their respective statements are presented in Table 4. Subsequently, axial, and selective qualitative structures were used to analyze the data on green logistics performance. Finally, the data were applied to the theoretical saturation test. The results of this analysis are presented in Table 5.

**Table 4.** Open coding analysis.

| Categories | Original Statements |
|---|---|
| 1. Current green logistics in urban tourism | Transportation is the main component of urban tourism that provides accessibility to tourist flows. International airports strongly support the global campaign for "sustainable aviation fuel (SAF)," and the priority slot will be given to airlines that operate with SAF and land at international airports. Private transportation enterprises have also introduced EVs to serve tourists in the EEC areas. Furthermore, tourists are encouraged to use bicycles to explore during their trips. Consumption and production are crucial components of green logistics; therefore, hotels in the EEC encourage tourists to reuse amenities during their stay. |
| 2. Greener goals | For the acceleration of green logistics to support greener goals, the government should prioritize the main objective, which is the fundamental reduction of environmental pollution and energy consumption through the implementation of green logistics and waste management policies. Green development goals in the EEC area are concerned with the large industry and tourism aspects at the macro level. Hotel businesses in the EEC area have implemented greener activities, for example, the use of Light Emitting Diodes (LED) in the hotel to reduce energy and be cost-effective. The hotel general manager stated that the rooftop hotel was constructed using a transparent ceiling to utilize the solar light during the daytime and the lights are turned on just after 6 p.m. to reduce energy consumption. |
| 3. Environmental impacts | The EEC area has large industrial operations, which cause huge environmental impacts due to the carbon emissions rate. However, the EEC is part of the economic strategy of the entire country. Tourism can play a vital role in initiating green logistics activities to reduce environmental impacts. Travel agents have created bicycle tourism to encourage tourists to use bicycles more often during their trips. Waste management is applied to reduce the impacts on the environment from hotels, by reusing towels, bringing one's mugs, and becoming paperless. |
| 4. Carbon emissions | One interviewee from the international airport stated that there is a policy to support sustainable aviation fuel (SAF) use for the carriers operating aircraft with SAF to have priority landing at the airport and a limitation of the size of aircraft flying in and out of the airport. Private transport companies emphasize using EV cars to transfer tourists from the airport to the hotel, utilizing the transportation as much as possible to reduce usage. Separating and reducing waste at the hotels is another potential method to reduce carbon emissions from consumer products or tourist stays at the hotels. |
| 5. Green design strategy | The concept of green design is quite vague depending on the industry and how this concept is applied. An airport that uses environmentally friendly construction and natural solar light might be promoted as a green airport. Hotel businesses can do more with the concept as many green concept hotel initiatives exist. Some restaurants use green design materials, for example, recycled paperware and straws. The green design concept can also be applied to brand marketing purposes. |
| 6. Transportation and Electric Vehicles options | Travel agents in the EEC have launched bicycle tourism and walkability campaigns to encourage tourists to use less transportation to reduce carbon emissions. Private transport companies also emphasize using EV cars to transfer tourists between airports and hotels. Private transport companies require support from the government to be diligent regarding eco-label vehicle practices, particularly in the EEC region, which has large transportation operations. |
| 7. Reverse logistics | The example of reusing towels and using mugs while staying at the hotel is applicable to reverse logistics. The policy of using recycled materials in restaurants must be implemented among all logistics enterprises involved in urban tourism activities in the EEC areas. |
| 8. The BCG Economy model to support green logistics performance | The BCG Economy model has been introduced by the Thai government as a tool to achieve economic growth and development goals; however, logistics enterprises in the EEC areas have little know-how about its adoption into practice although bio-diesel fuel has been used nationwide to encourage consumers to use it for their transportation. It must be properly implemented in the EEC areas to set the best practices for using the BCG Economy model in urban tourism activities. |

**Table 4.** *Cont.*

| | Categories | Original Statements |
|---|---|---|
| 9. | Reduction of GHG and carbon emissions | The BCG Economy model is one of the key drivers in reducing GHG and carbon emissions. If the proper policy is launched, according to the deputy director of the international airport, the transportation side will be willing to respond to the policy, while hotel owners will try to reduce waste in the hotel generated by tourist stays. Bicycle tourism and walkability programs should be promoted by travel agents to their customers (tourists). |
| 10. | Reduction of energy consumption and usage | LED lighting has been installed in the hotels, and a transparent glass ceiling has been fitted on the rooftop to use more natural solar energy light during the day and use electricity only at night. The international airport also uses all LED lights to reduce energy usage and be cost-effective, which is proven by comparing energy consumption before and after changing to the LED. |
| 11. | Government intervention and policy | Private transportation owners recommend that low carbon emissions taxation should be legally implemented to encourage the use of EV vehicles, and hotels, restaurants, travel agents, and destinations expressed that the effective lower taxation for enterprises that can reduce the carbon emissions rate must be a proper policy because transforming logistics to green logistics requires a high level of investment, as the green products have high added value and high prices. |
| 12. | Business sustainability | The international airport in the EEC emphasizes its sustainability mission and does all that is possible to achieve sustainability. Hotel businesses support the use of bottles of water produced by low carbon footprint materials in the hotel. Green logistics can also support their business sustainability. |
| 13. | Development of the entire logistics industry | Enhancement of the development of the entire logistics industry requires technology to support efficient infrastructure. The proper serious support from government policy, particularly in the EEC region, needs to leverage the development of logistics in the region. This will contribute to the urban tourism perspective when the entire logistics system is developed. |

**Table 5.** Axial and Selective Coding.

| Main Categories | Corresponding Categories | Connotation Categories |
|---|---|---|
| Implementation of the green transportation system | Transportation and Electric Vehicles (EVs) options Current green logistics in urban tourism | Logistics in urban tourism is related to transportation providing accessibility to reach the destination; thus, transportation is the main component required by consumers that are here as tourists. Hence, green transportation is also the main part of green logistics performance. The international airport in the EEC attempts to reduce energy usage at the airport, supports airlines that fly with SAF, and utilizes more EV cars and buses in the areas for tourist facilitation. Bicycle tourism and walkability lanes have also been implemented for tourists. |
| Enhancement of reverse logistics | Reverse logistics Reduced energy usage and consumption | Reverse logistics involves the consumption process from production to the end-consumer. It refers to reusing, recycling, and reducing, which are the main components of reverse logistics, and also the indicators of green logistics performance. In terms of urban tourism, the hotels can contribute by supporting the reuse of amenities such as towels, mugs, and bedding, the use of recycled products in the hotels, and reducing the use of energy, such as establishing rooms that can have solar light usage instead of fully depending on electric lighting. |

**Table 5.** *Cont.*

| Main Categories | Corresponding Categories | Connotation Categories |
|---|---|---|
| The level of the environmental management system | Environmental impacts Carbon emissions Reduced GHG and carbon emissions | The effectiveness of green logistics performance initiated by local logistics enterprises is vital for environmental management systems to reduce the carbon emissions rate and reducing energy usage supports the environmental management system. Walkability and bicycle tourism initiatives through travel and tourism enterprises reduce GHG and carbon emissions rates. |
| The level of governance of the government | Greener goals BCG Economy model Government Intervention and Policy | The intensity of the implementation of green policy, the environmental regulations capacity of green policy, and the coordination of related departments of green governance affect the green governance capacity of the government. The proper policy of taxation reduction in terms of carbon emissions rates must be implemented to encourage enterprises to pay more attention to greener goals for green logistics performance. Moreover, those involved must be properly educated about the Bio-Circular-Green Economy model and the tools needed to achieve sustainable development. |
| The perceived usefulness of green logistics performance by the green logistics enterprises | Green design strategy Business sustainability Development of the entire logistics industry | Green logistics performance is affected by green logistics enterprises; hence, green design strategy should be included in the enterprises' business plans and can also benefit long-term business sustainability. Subsequently, the development of the entire logistics industry in the EEC region will impact the perspectives of the logistics enterprises in urban tourism to be more aware of green logistic performance. |

After 13 categories were formed during open coding, the relationship between those categories was divided into five main categories with correspondence using the re-analyzed data shown in Table 5. The result of the green logistics performance assessment is affected by the logistics enterprises' perception of the usefulness and ease of use of green logistics performance. The data were then analyzed and compared for the effectiveness of the green logistics performance (GLP) to respond to the research objective of this study, as seen in Table 4. It was found that the effectiveness of GLP from an urban tourism perspective is affected by the green transportation system, reverse logistics, environmental management system, GLP's achievement of sustainable development, and the government's level of governance. Table 5 shows the relationships between the open coding of green logistics performance in urban tourism, which was obtained through interviews with logistics enterprises. Therefore, the connotation categories of green logistics performance are presented in the axial coding.

*4.4. The Final Steps Consolidating the Result*

The results of the study indicated that the effectiveness of GLP, from the urban tourism perspective in the EEC region, is affected by the green transportation system, reverse logistics, the environmental management system, the level of governance of the government, and GLP's achievement of sustainable development through green logistics enterprises. Green transportation is fundamental for the effectiveness of green logistics in the urban tourism sector as it facilitates tourist flows when moving from one place to another during their travels and is the main component of accessibility for tourism destination development. Bicycle tourism and walkability activities for tourists support effective green logistics, international airports should support the global sustainable aviation fuel campaign, and private transportation is focusing more on EVs and low-carbon fuel. The GLP is affected by environmental management systems. Reverse logistics, which consists of reusing, reducing, and recycling products in consumption and production, is the main component of green

logistics and impacts the contribution of green logistics performance. The government's governance level supports the contribution of green logistics performance. The strength of the governance capacity of the government affects policy implementation. If the governance capacity is weak, it may be because the government does not pay sufficient attention to greener goals. The proper policy of a carbon offset encourages logistics enterprises to operate their businesses as usual while reducing the GHG effects that are caused by their business. Green logistics enterprises are encouraged by pure green policies based on the government's level of governance about the initiation of any green logistics activities from an urban tourism perspective. These greener goal initiatives need to be supported by the government for them to be recognized by the enterprises.

The advantage of the development level of the logistics industry is the emphasis on the environment for GLP operations and the development of logistics enterprises. The higher the level of development of the logistics industry, the greater the value of the GLP; thus, the effectiveness of the GLP is also affected. Therefore, although the development level of the logistics industry also does not directly affect the performance of green logistics, it will affect the intensity of the contribution of GLP. Thus, the development level of the entire logistics industry affects the logistics enterprises regarding GLP and the achievement of sustainable development.

The result of the theoretical saturation test found that no new and important category was formed, as all of them remain the same as when based on the data from the first key informants' interviews. Likewise, the five interviewees did not provide any new critical views. They indicated that these are the factors that influence the effectiveness of green logistics performance. Following the saturation test, the proposed model of contributions of green logistics performance and the factors that potentially influence the contribution of green logistics performance were established in Figure 3.

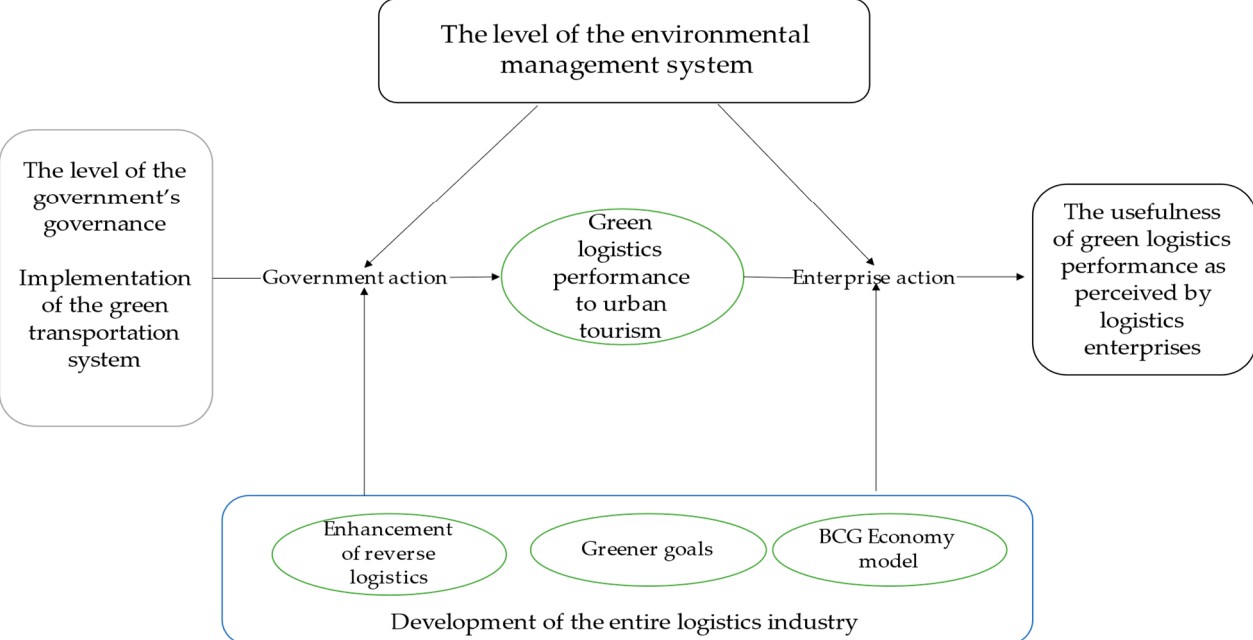

**Figure 3.** The model of contributions of green logistic performance in urban tourism. Source: Developed by authors.

## 5. Discussion

The results indicate that five factors impact green logistics performance (GLP) assessment: The implementation of green transportation, the enhancement of reverse logistics, the level of the environmental management system, the level of the government's governance, and the usefulness of GLP as perceived by green logistics enterprises. The findings are integrated with the research objective to assess the current performance of green logistics

in urban tourism. Consequently, this paper's proposed GLP influencing factors model confirmed the previous studies and theories on GLP. First, Zhang et al. 's model of the green logistics policy implementation process was developed for GLP policy implementation recommendations. Zhang et al. (2020) argue that idealized policy, implementing organizations, target groups, and environmental factors are important, similar to the view of this study.

### 5.1. Implementation of the Green Transportation System

Based on the logistics performance index (LPI), transportation is a primary indicator of logistics performance; therefore, green transportation systems in urban tourism play a vital role in GLP as it facilitates the flow of tourists. The axial and selective coding results support the study of Larina et al. (2021) on a green transportation system's reduction of GHG and carbon emissions. Green transportation systems can implement sustainable aviation fuel (SAF), use more EV vehicles, and promote a greener environment. Bicycle tourism and walkability are effective environmentally friendly tourism approaches, similar to Rezvani Ghomi et al. (2021).

### 5.2. Enhancement of Reverse Logistics

Reverse logistics is a key factor to assess green logistics' potential to achieve sustainable urban tourism development, which confirmed the study of Pourmehdi et al. (2022), who stated that reverse logistics consists of the 3 Rs of reusing, reducing, and recycling, similar to the reverse logistics factors to assess green logistics in urban tourism. Hotels can encourage guests to reuse utensils such as mugs during their stay. Travel agents can launch bicycle tourism and walkability programs to reduce vehicle use and minimize the carbon emissions rate. Restaurants can help minimize the $CO_2$ footprint from the bottles of water served by supporting the production of low-$CO_2$-footprint bio bottles, each of which saves 56 g of $CO_2$ emissions. Energy usage in hotels can be reduced by using LED and solar lighting.

### 5.3. The Level of the Environmental Management System

Tourism has major environmental impacts, and tourists are defined as consumers from the tourism industry perspective who consume various logistics activities. This study is in line with Testa et al. (2021), who stated that environmental protection and emissions reductions should be considered by tourism industry enterprises. Logistics enterprises and government bureaus have revealed the importance of environmental management in logistics activities in urban tourism. Green goals can provide systematic environmental management from enterprises to ensure support from government policy. In climate change mitigation, it is a challenge for governments to reduce $CO_2$ emissions from logistics and transportation; hence, the environmental management system is a major factor for GLP in this research and can improve tourists' well-being regarding social aspects. When tourist numbers increase, it contributes to long-term sustainable economic development.

### 5.4. The Level of Governance of the Government

In this research, the level of government governance refers to its ability to achieve the goal of green logistics, including the intensity of the implementation of green policy, environmental regulation capacity, promotional capacity, and coordination of related departments. In the long term, the development of green logistics requires huge investments from both the government and private enterprises, whereas the interchangeability of the mutual benefits between the government and enterprises must be agreed upon. McKinnon (McKinnon 2021) confirmed that policymakers do not always support policy implementation with the same strength. For example, in Thailand's efforts to clean up pollution, many local governments do not realize the problem's importance and continue to approve projects that cause pollution. Enterprises also do not realize the significance of businesses' impacts on the environment. These local and central governments' low green governance capacity leads to low GLP.

*5.5. Perceived Usefulness of GLP for Green Logistics Enterprises*

Perceived usefulness is the extent to which a person believes that the use of a system will enhance job performance, while perceived ease of use is the extent to which a person believes that using a particular system will be free of effort. Similarly, previous research identified that the perception of enterprises is a crucial factor in implementing green logistics performance, this research found that the level of the perception of logistics enterprises regarding GLP affects their degree of acceptance of the policy. When logistics enterprises realize the usefulness of GLP, their willingness to accept and implement it in their business activities is enhanced, thereby improving the effectiveness of the GLP.

## 6. Conclusions

The concept of green logistics should be vigorously promoted to improve the quality of the environment, save energy, and reduce emissions of logistics enterprises, especially those involved in urban tourism in the EEC region. Green logistics and sustainable tourism are closely intertwined concepts, complementing each other to preserve a high-quality and clean environment as the first priority, followed by improving the well-being of people in society, including tourists and residents who can benefit from GLP. Green logistics performance also contributes to economic aspects when attempting to minimize resource usage by adopting the Bio-Circular-Green Economy (BCG) model to maximize resources produced in green logistics activities. Likewise, GLP complements sustainable development in the aspects: Environment, Society, and Economics. Nonetheless, the challenges and opportunities of GLP depend on the level of governance of the government.

These findings make significant contributions to the managers and administrators involved in Green Logistics Policy (GLP). Firstly, the established model framework can help policymakers prioritize factors essential to GLP's success. This will support governments in developing GLP. Secondly, the government must prioritize educating, supervising, and enhancing the capacity and efficiency of green governance. Thirdly, while implementing GLP, the government should improve the policy system by considering the current development level of the entire logistics. Additionally, this research provides valuable insights for logistics enterprises in the EEC region. Based on these findings, they can incorporate green concepts into their strategic business plans, promote green behavior, increase visitors' awareness, and engage in corporate social responsibility (CSR) projects. Non-green behavior by logistics enterprises has significant negative impacts on the environment. Hence, it is crucial for businesses to acknowledge the severity of the issue and comply with the many guidelines and regulations introduced by the government to enhance green behavior.

There are two limitations to this research that need to be acknowledged. Firstly, this study focuses on green logistics from a tourism perspective, which means that it does not cover the entire logistics industry. Secondly, the data were collected from the EEC region of Thailand, and the theoretical framework developed may not be applicable to the entire country. Therefore, future research should explore different areas and perspectives to collect further data and verify the theoretical model through quantitative methods. In summary, while this study provides valuable insights into green logistics policies in the EEC region, its findings may not be generalizable to other regions or industries. Further research is necessary to develop a more comprehensive understanding of green logistics policies in Thailand.

**Author Contributions:** The authors contributed equally to this work. Conceptualization, S.V., V.N., N.S. and D.S.S.; Methodology, S.V., V.N., N.S. and D.S.S.; Writing—original draft preparation, S.V., V.N., N.S. and D.S.S.; writing—review and editing, S.V., V.N., N.S. and D.S.S. All authors have read and agreed to the published version of the manuscript.

**Funding:** This research received no external funding.

**Institutional Review Board Statement:** The study was conducted in accordance with the Institutional Review Board for Ethics Committee of Asian Institute of Technology code RERC 2022/013, 29 March 2023.

**Informed Consent Statement:** Informed consent was obtained from all subjects involved in the study.

**Data Availability Statement:** Some data in this study are publicly available in World Development Indicators published by World Bank (2019) at https://databank.worldbank.org/source/world-development-indicators (accessed on 23 October 2022).

**Conflicts of Interest:** The authors declare no conflict of interest.

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
