# Peer review of "Assessment of the Factors Influencing the Performance of the Adoption of Green Logistics in Urban Tourism in Thailand’s Eastern Economic Corridor"

_socsci, doi:10.3390/socsci12050300_

Round 1

Reviewer 1 Report

Dear authors, 

Congratulations for a good work, well worked.

For me, your paper is ok but, perhaps, it lacks of some more numeric arguments (like LPI possible values). Please, try to work this aspect for future papers.

Only one little typesetting mistake; page 16. Figure 2 should be figure 3

Author Response

Response to Reviewer 1 Comments

Congratulations for a good work, well worked.

For me, your paper is ok but, perhaps, it lacks of some more numeric arguments (like LPI possible values). Please, try to work this aspect for future papers. Only one little typesetting mistake; page 16. Figure 2 should be figure 3

Response 1: Please provide your response for Point 1.

Thank you for the comment. Adding the numeric arguments related to LPI possible value “McKinnon stated that the logistics performance index (LPI) is one of the indicators to assess the performance of logistics, according to the logistics performance index (LPI) certified by the world bank in 2018, Thailand is ranked no.34 out of 167, highest performance reached 80.20% (Li and Muhammad 2021), the number indicates efficiently operating of logistics which means the higher efficiency is higher impact environment. Hence, this is to reinforce that logistics performance in Thailand is necessary for the transformation toward green logistics (page 5/20)

Response 2: Please provide your response for Point 2.

Thank you for the comment. Figure 2 has been revised to 3 on (page 16/20)

Reviewer 2 Report

The study is interesting, but the sample size is inadequate. Although the study is context specific.  At this present form, the paper should be considered after minor spelling checks. 

Author Response

Response to Reviewer 2 Comments

Comments and Suggestions for Authors

The study is interesting, but the sample size is inadequate. Although the study is context specific.  At this present form, the paper should be considered after minor spelling checks. 

Response 1: Please provide your response for Point 1.

Thank you for the comment. We acknowledged your comment and would state that “the sampling size in qualitative research, Creswell and Cresswell (2017) suggested that the number of interviews should be between 20 to 30 for reliable results”. Also, the study area is specific to logistics enterprises from an urban tourism perspective hence the number of 25 logistics enterprises in the urban tourism area is considered sufficient for this study. And thank you for the suggestion in the next study we will be more careful with a larger sample size (page 8/20)

Reviewer 3 Report

Abstract: 

- Include the underlying contribution from the findings, and what will be the methods applied in the research. 

Introduction

- Last para: that is not the problem statement... where the problem statement is still unclear

Literature Review: 

- underpinning theories should be highlighted in early LR

- the conceptual and significant explanation on sustainable development with the green logistic should be explained further. 

- research methodology 

- The method developed is not clear and what will the method going to apply

- Discussion could be more thoroughly discuss and explain which integrated the findings, and answering the research duit. 

Author Response

Response to Reviewer 3 Comments

Comments and Suggestions for Authors

Abstract: 

- Include the underlying contribution from the findings, and what will be the methods applied in the research. 

Response 1: Please provide your response for Point 1.

Thank you for your comment. We acknowledged that and added the clear methods applied in the research stated “The findings based on in-depth interviews with logistics enterprises that the contributions of green logistics performance to urban tourism with a holistic perspective imply academic and can provide the most recent developments for better-informed decisions regarding the enhancement and improvement of green logistics performance in terms of urban tourism and sustainable regional development

The methodology used a qualitatively semi-structured interview approach with the logistics enterprise

(page 1/20)

Introduction

- Last para: that is not the problem statement... where the problem statement is still unclear

Response 2: Please provide your response for Point 2.

Thank you for the comment. We acknowledged that

Effective green logistics performance is the ultimate goal of the EEC development plan for this regional gateway of tourism growth. Logistics enterprises have played a significant role in adopting green logistics for urban tourism. However, there is still a gap, as many logistics enterprises are not focused on green logistics. To address this gap, this study aims to assess the factors influencing the performance of the adoption of green logistics in urban tourism in the EEC area of Thailand, which could form a basis for better-informed decision-making toward sustainable development. (Page 2-3/20)

Literature Review: 

- underpinning theories should be highlighted in early LR

Response 3: Please provide your response for Point 3.

Thank you for the comment. We acknowledged that and restructure the theories in early LR as The interlink between green logistics and urban tourism is clear and can be seen in the green transport-related activities and the entire infrastructure that play a role in the accessibility of tourists to a certain destination, which has a direct impact on the efficiency of sustainable tourism, as well as fuel-efficient vehicles, biofuels, electric vehicles (EVs), bicycles, and tricycles for delivery (Rashidi and Cullinane 2019). Other concerns are the consumption and production of green logistics in urban tourism areas, including accommodation, restaurants, and sightseeing in the entire package provided by the enterprises to tourists (page 3/20)

- the conceptual and significant explanation on sustainable development with the green logistic should be explained further. 

Response 4: Please provide your response for Point 4.

Thank you for the comment. We acknowledged that to further explain the concept of green logistics contributes to sustainable development states “The contribution of the logistics sector considerably impacts the growth of the country’s economy, which involves several sectors, such as transportation, consumption, and production in the general industries related to the logistics sector. However, logistics from the tourism perspective concerns transportation, accommodation, restaurants, attractions at tourist destinations, etc. The UNWTO defines sustainable tourism as “tourism that yields current and future economic, environmental, and social impacts on the needs of the industry, visitors, host communities, and the environment.” Thereby, the “green” or “sustainable” concept advocates for minimal detrimental impacts on the local environment and the promotion of cultural benefits in the region for its local inhabitants in the social pillar of the SDGs” (page 5/20)

Research methodology 

- The method developed is not clear and what will the method going to apply

Response 5: Please provide your response for Point 5.

Thank you for the comment. We acknowledged that to state clearly the method applies “Based on the literature, the qualitative method of semi-structured interviews conducted self-administered through face-to-face, Zoom conference, and telephone, with logistics enterprise as an appropriate approach to reach the assessment factor of green logistics performance in urban tourism contributing to sustainable development by the logistics enterprises in the Eastern Economic Corridor of Thailand. Thus, the data were recorded by memo and voice recording to transcript it for data analysis” (page 6/20)

- Discussion could be more thoroughly discuss and explain which integrated the findings, and answering the research duit. 

Response 6: Please provide your response for Point 6.

Thank you for the comment. We acknowledged that to more thoroughly explain the integration of findings as stated “The results indicate that five factors impact green logistics performance (GLP) assessment: implementation of green transportation, enhancement of reverse logistics, the level of the environmental management system, the level of the government’s governance, and the usefulness of GLP as perceived by green logistics enterprises. The findings are integrated with the research objective to assess the current performance of green logistics in urban tourism. Consequently, this paper's proposed GLP influencing factors model confirmed the previous studies and theories on GLP. First, Zhang et al. ‘s model of the green logistics policy implementation process was developed for GLP policy implementation recommendations. Zhang et al. (2020) argue that idealized policy, implementing organizations, target groups and environmental factors are important, similar to the view of this study. (Page 16/20)

Reviewer 4 Report

The authors deal with a very important topic and the beginning of their work looks very promising.

However, gaps and shortcomings are found in the development of the discussion:

- the authors intend to deepen the theme of urban tourism, but the description of the area analyzed seems to include other types of tourism as well, given that seaside attractions are mentioned. Table 1 describes the three provinces: it is not clear whether tourist arrivals refer only to urban contexts or whether they also include seaside area

- still with regard to urban tourism, no element is given that could make us understand the articulation between business tourism, cultural tourism, hub tourism, etc., targets with very different travel and stay behavior. These are important aspects for establishing green logistics measures. The authors should explain this issue

- the theme of green logistics, as the authors also argue, is complex and affects all aspects of mobility and the many types of use of the urban context: citizens, students, workers, commuters, even before tourism. Particularly for transport and mobility, needs can be partially overlapping, but in many cases they differ depending on the target being considered. This would mean, on the one hand, highlighting the aspects common to the various targets and which would in any case have a positive impact on tourism (from electric cars to the use of bicycles, etc.), on the other, identifying those specific to tourists (business? cultural tourists?).  The authors should deepen these aspects, both in terms of the literature and the description of the study area

- although a qualitative analysis does not actually require a large sample size, however it would be important, especially for the interviewees such as hotels, travel agencies, transport companies, to indicate how much they represent of the entire market and specify if they have produced official documents regarding the declared green strategies

Author Response

Response to Reviewer 4 Comments

Comments and Suggestions for Authors

The authors deal with a very important topic and the beginning of their work looks very promising.

However, gaps and shortcomings are found in the development of the discussion:

- the authors intend to deepen the theme of urban tourism, but the description of the area analyzed seems to include other types of tourism as well, given that seaside attractions are mentioned. Table 1 describes the three provinces: it is not clear whether tourist arrivals refer only to urban contexts or whether they also include seaside area

Response 1: Please provide your response for Point 1.

Thank you for the comment. We acknowledged that this study specifically focuses on the urban tourism context therefore, Table 1 describes the three provinces in the region of Eastern Economic Corridor (EEC) which must include three provinces and study focuses on tourists’ arrival in the urban area context where the data collection has conducted not other types of tourism. The terms seaside has been mentioned for the reason that this area has other attractions however, we agreed with the comments that focus on urban tourism then should be only unbanned context thus the term seaside has been removed (page 7/20)

- still with regard to urban tourism, no element is given that could make us understand the articulation between business tourism, cultural tourism, hub tourism, etc., targets with very different travel and stay behavior. These are important aspects for establishing green logistics measures. The authors should explain this issue

Response 2: Please provide your response for Point 2.

Thank you for the comment. We acknowledged that and more explanation is given in regard to urban tourism. Adding the paragraph to explain more as stated “The United Nations World Tourism Organization (UNWTO) defined urban tourism as “a tourist activity in an urban area, while city/urban destinations provide a wide range and variety of attractions in cultural, architectural, technological, social and natural experiences and products for leisure and business.” Urban tourism planning and management require the equality of the logistics system to develop at the same time with the conservation of the environment, culture, and society, as well as transportation within the urban areas, to support sustainable development (Han 2021). (page 4/20)

- the theme of green logistics, as the authors also argue, is complex and affects all aspects of mobility and the many types of use of the urban context: citizens, students, workers, commuters, even before tourism. Particularly for transport and mobility, needs can be partially overlapping, but in many cases they differ depending on the target being considered. This would mean, on the one hand, highlighting the aspects common to the various targets and which would in any case have a positive impact on tourism (from electric cars to the use of bicycles, etc.), on the other, identifying those specific to tourists (business? cultural tourists?).  The authors should deepen these aspects, both in terms of the literature and the description of the study area

Response 3: Please provide your response for Point 3.

Thank you for the comment. We acknowledged that explain more about green logistics use in the urban tourism context where tourists primarily involve such as transport, mobility, consumption and production adding in the literature part and description clearer in the study area part as stated “green logistics performance are integrated to manage the flow of humankind and have an essential role in urban tourism. It has been proven that green logistics are a key contributor to sustainable tourism development as green logistics systems in tourism provide facilities for tourists to reach their destination during their stay. (Muangpan and Suthiwartnarueput 2019; McKinnon 2021) defined logistics in tourism perspectives as involving transportation and composed of the carriers, accommodation places, restaurants, locations for sightseeing, man-made attractions, car rental firms and the whole setting, including the décor, appearance of staff, and timeliness of the service received, for which it is essential to have co-operation and the co-ordination of the different activities and areas (page 2-3/20)

The study area covers Thailand's Eastern Economic Corridor (EEC) region (Table 1 and Figure 2). The EEC is a diverse area that offers tourists and visitors a wide range of attractions. The region consists of three provinces (Chachoengsao, Chonburi, and Rayong), with several famous attractions in the urban area. Each province has its characteristics in terms of urban tourism perspectives and involvement with logistics enterprises. Green logistics performance in the study area involves transport-related, international airport, accommodation, restaurant, and tourist attraction destinations (page 7/20)

- although a qualitative analysis does not actually require a large sample size, however it would be important, especially for the interviewees such as hotels, travel agencies, transport companies, to indicate how much they represent of the entire market and specify if they have produced official documents regarding the declared green strategies

Response 4: Please provide your response for Point 4.

Thank you for the comment. We acknowledged that to define the sampling size in qualitative we use “the sampling size in qualitative research, Creswell and Cresswell (2017) suggested that the number of interviews should be between 20 to 30 for reliable results”. Also, the study area is specific to logistics enterprises from an urban tourism perspective hence the number of 25 logistics enterprises in the urban tourism area is sufficient for this study. And thank you for the suggestion in the next study we will be more careful with a larger sample size (page 8/20)

Further, the classification, as the study covers three provinces and logistics enterprises are selected for sampling size  total of 25 enterprises from different fields here shows how much they represent of the entire market

Province

Government/Logistics Enterprise

Position

Gender

No.

Entire Market

Chachoengsao

Hotel

Owner

Male

2

17

Chonburi

Private transport

Hotel

Restaurant

Travel agency

Owner

HR Director

Owner

Owner

Male

Female

Male/Female

Male

3

2

4

2

10

500 +

N/A

8

Rayong

International Airport

Government bureau

Private transport

Hotel

Restaurant

Destination

Deputy Director

Director

Owner

GM

Owner

Manager

Male

Male

Male

Male

Male/Female

Male

2

1

2

1

4

2

1

1

10

125

N/A

4

Total

25

**sources: The Eastern Economic Corridor Office and Agoda Hotels and Attractions website

*** N/A The data is not available due to during covid some restaurant has closed down

specify if they have produced official documents regarding the declared green strategies

Response 5: Please provide your response for Point 4.

Thank you for the comment. We acknowledged that the green concept/strategies and sustainable development have been officially a policy from the Eastern Economic Corridor Office of Thailand, it is a government bureau agency to govern the entire region of ECC which consists of three provinces which are the study areas in this research. Green strategy from the EEC office involves the whole industry in the EEC including the tourism part.

Another official is from U-Ta-Pao International Airport where the interviewee as sampling has been interviewed, the airport has launched, a green space concept, Ecological, Remidiaiton, Community Environment, and Water retention policy using a green strategy to initiate green logistics (page 7/20)

Round 2

Reviewer 4 Report

Thank you so much to the authors. They have covered all the required additions and changes